# Usage of Evolutionary Algorithms in Swarm Robotics and Design Problems

**DOI:** 10.3390/s22124437

**Published:** 2022-06-11

**Authors:** Levent Türkler, Taner Akkan, Lütfiye Özlem Akkan

**Affiliations:** 1The Graduate School of Natural and Applied Sciences, Dokuz Eylul University, Buca, İzmir 35390, Turkey; 2İzmir Vocational School, Dokuz Eylul University, Buca, İzmir 35380, Turkey; taner.akkan@deu.edu.tr (T.A.); ozlem.karaca@deu.edu.tr (L.Ö.A.)

**Keywords:** swarm robotics, optimization, multi robotic systems, evolutionary algorithm, simulation

## Abstract

In this study, the general structure of swarm robotics is examined. Algorithms inspired by nature, which form the basis of swarm robotics, are introduced. Communication topologies in robotic swarms, which are similar to the communication methods between living things moving in nature, are included and how these can be used in swarm communication is emphasized. With the developed algorithms, how the swarm can imitate nature and what tasks it can perform have been explained. The various problems that will be encountered in terms of the design of the optimization methods used during the control of the swarm and the solutions are simulated using the Webots software. As a result, ideas on the solutions of these problems and suggestions are proposed.

## 1. Introduction

Robot cooperation to perform tasks for a specific objective is widely studied, using the concept of multi-robot. This unified work is achieved by autonomous structure and good communication [1]. Robots must solve the encountered problems individually on their own. The movements of the swarm robots therefore should be autonomous rather than centrally managed.

Generally, swarm robots are smaller and less functional robots compared to other types of robots. The function of the robot swarm is directly related to successful task performance by the individual members of the swarm. Although the increase in the number of robots and the reduction of their structural size give the impression that the success rate in their tasks may decrease, in fact the power of collective work actually increases. The robots in the swarm are designed to overcome problems more effectively than single robots dealing with the same problems. For example, on a larger, more powerful robot, a system failure could risk the mission on a large scale. However, in most cases, it is not important if one or more robots are disabled in the robot swarm and the task can still be completed successfully. Moreover, successful autonomous work may require a number of sensors. Structurally simple swarm robots might be thought to have too few sensors, but a swarm of thousands of individuals naturally has thousands of sensors.

The sensorial concept and sensor problems constitute an important research area in swarm robotics. While algorithms are valid for the prediction of swarm movement at the macroscopic level, the dynamics of the components of the swarm and their interactions with their environment come to the fore at the microscopic level [2]. The robot environment and interactions required by the algorithms are realized by transmitting the data from the sensors. Therefore, sensors play a very important role in swarm robotics. Especially in unknown environments, sensors are required for the ability of robots to perform safe and efficient movements [3]. Sometimes sensors prevent collisions with other robots, sometimes they help in exploring the environment, and sometimes they perform tasks such as providing information about the locations of the robots [4]. For example, Stirling et al., in their study, did not find it appropriate to detect location information with the Global Information System due to the high error rates in mobile systems with many robots [5]. However, they have shown that robots with different sensors, such as Relative Localization Sensor, can provide faster and more accurate results in swarm applications [6]. Another study proposes the control of swarm robotics in order to perform challenging tasks [7]. Here, exploratory mission simulation was carried out with proximity and color sensors in the region thought to contain radiation. In another study, it is stated that the foundations of swarm intelligence are formed by individuals thanks to sensors [8]. The study aimed to create the necessary conditions for autonomous movement by collecting the information from the sensors on the swarm robots in the field. An autonomous organism formation is sought with the help of these sensors and therefore the information coming from individuals. Additionally, using more complex sensors offers real-life solutions. For example, it is currently known that efficient solutions are provided with high resolution cameras and high-speed communication networks in agricultural areas by using unmanned aerial vehicles (UAVs). Lin Shi et al. presented a study on this subject [9].

Controlling many robots, sometimes thousands, is more difficult than controlling a single robot. It is almost impossible to control these robot swarms one by one. For this reason, it is a must for robots to be autonomous in swarm robotics and solve the problem on their own. When the user sends the task to the swarm, the swarm must exhibit the pre-defined behavior to perform the given task. In some cases, the user and/or the central system may interfere with the swarm. However, this intervention does not involve the control of a single robot, but the whole or a large part of the swarm. Here, giving information to the whole swarm or enabling the swarm to do the same work together requires very strong communication.

If the swarm robots are self-moving and the control has more decentralized architecture, nature is the best source of inspiration to provide the working logic of these robots. Many living species in nature move in groups and often come together to fulfill a task. Swarm control is achieved again by the swarms controlling each other. It is not a correct point of view to look at this phenomenon as central management or leadership. Providing the best movement for the group is achieved by imitating the individual in the most suitable position by all members of the group [10]. Is it possible to transfer this ability of living things or nature to swarm robotics? There are many studies on this subject. Influenced by nature, swarm robots will fulfill their duties jointly as is seen in nature. While performing their duties, perhaps it would be more beneficial to act by collecting the behavior of several different living species in a robot. In this sense, it is necessary to examine the living things that move together in nature and the optimization methods adapted to this.

There are different studies on evolutionary optimization techniques of swarm robotics. One research is about three-dimensional route planning using Beetle Swarm Optimization studied by Yizhuo Mu et al. [11]. The three-dimensional space environments such as air and underwater are not suitable for traditional route planning of robots because of their nonstructured models. Therefore, swarm robot optimization methods are suitable due to their intelligence that make easy to deal with the complex environmental structures. This intelligence naturally deals with nonlinearity and produces stable solutions. In another research, Jiajie Liu et al. studied multi-objective evolutionary optimization methods to prioritize multiple tasks for Unmanned Aerial Vehicles (UAVs) [12]. Similarly for UAVs, Akshya et al. studied area partitioning, considering static obstacles and altitude parameters using Firefly and Particle Swarm Optimization (PSO) algorithms [13]. Another study is about optimum route navigation and collision avoidance of wall follower robots using six different evolutionary optimization methods as PSO, Multi-Verse Optimizer (MVO), moth-flame optimization (MFO), cuckoo search (CS), grey wolf optimizer (GWO) and bat algorithm. Here, several ultrasonic sensors mounted on the robot are utilized to detect objects studied by Jalali et al. [3]. In another research, a novel dynamic cooperative co-evolving PSO algorithm, derived from an evolutionary algorithm, was used and studied for multi-robot formations by Lee et al. [14]. Another study is about an evolutionary optimization method for effective route planning of drone swarms developed by Majd et al. [15]. In another research, bio-inspired and evolutionary algorithms were used to control a self-reconfigurable modular robot platform in swarm mode. The developed simulator assigns robots of a swarm to form an organism-like behavior for locomotion tasks. Instead of developing hardware robots, this approach saves time through developing an algorithm using simulation environment. This study is conducted by Winkler et al. [16]. Another study is about drone path planning using lion swarm hybrid differential evolutionary algorithm made by Liu et al. Lastly, different from swarm robotic applications, a medical imaging diagnostic system was studied for accelerating training speed and increasing accuracy using genetic and PSO algorithms by Chen et al. [17]. As can be seen from the last research example, these algorithms find wide application not only in swarm robotics but also in many other fields.

The harmony in nature is the result of a very long process of experience and is perfect. The tasks were settled among the swarm by time, and the adjustment to physical difficulties was transferred to the swarm. However, swarm robotics is still in its beginning stages. For this reason, problems in the selected algorithms, communication methods, and mechanical structures might be still encountered. These problems are generally tackled by looking at nature again or by using different mechanical methods.

Many application examples exist from the perspective of swarm robotics. For example, in an above-mentioned study swarm task optimization of UAVs was developed for a rescue environment. Here, evolutionary algorithms needed to overcome UAV swarm communication limitations and difficulties of rescue areas to avoid extensive computational loads [12]. Again, the sampled positions and trajectories of a swarm of five robots were examined for multirobot system control by Lee et al. [14]. Another study by Majd et al. was about drone swarms and collision prevention between drones and static or dynamic obstacles with route optimization [15]. In another study, UAV swarming was used for inventory counting using high precision RFID readers and their trajectory was planned by the lion swarm algorithm [18].

In this study, we will focus on how swarm robots communicate with each other, how we understand nature, how we move robots in robotic work areas using optimization techniques, the tasks of robots, design problems during the test, and their solutions. In the next section, general information will be given and in its sub-sections features of swarm robotics and their classification from different perspectives will be presented. Then the communication methods and topologies used in swarm robotics will be discussed. The following sub-sections show how swarm robotics takes nature as an example and the optimization methods will be explained and examples from real life in the literature will be cited. In section three, the design problems and proposed solutions will be discussed. Finally, the paper will conclude with the results of the proposed study.

## 2. Materials and Methods

### 2.1. Multi-Robot Systems

Multi-robots are a group of robots that emerged with the idea of multiple robots doing the same mission together. In robot communities, the fact that individuals do the same work does not always constitute the concept of a swarm. To call a group of robots a swarm there should be some distinguishing features of the robots such as the number, task, similarity, collaborative ability, and control technique. By looking at these distinctive features, it is decided whether a robot community is a swarm or not. In Table 1, the differences between the communities consisting of many robots are defined.

The parameters that differentiate these systems from each other are:Number of Elements: Number of agents, individuals in the robot community.Control Type: How the robot community’s movements are governed.Element Type: The similarity of community members to each other.Flexibility: Allowing community members to change behavior in response to environmental conditions.Robustness: The ability of the system to continue its function in failure or unexpected situations.Scalability: The effect of the reduction of the number of community members on the task.Environment: The status of the defined environment where the group will work.Usage areas: General usage area of the group [19].

As concluded here, although the concept of a robot is defined as machines with manipulators, in some cases, it is also possible to talk about virtual systems in multi-robot architecture. These systems are usually very large computational tasks running parallel in different processing units with the help of distributed tasks. 

Our field of study here is swarm robotics, which is a relatively new subject. In this sense, the expression “robot” in the texts from now on should be considered as individuals or agents of swarm robotics.

### 2.2. Swarm Robotic

The concept of swarm robotics (SR) is a division of multi-robot systems. Although there are very significant and varied studies in this field, the biggest feature that distinguishes them from multi-robots is their autonomous structure. This feature is both the strongest and the most important characteristic of swarm robotics. 

Individuals in the swarm can adapt themselves to different tasks in a wide variety of conditions, as can be deduced from the classification methods in Table 1. Working with autonomous systems was described as collaborative by Luca Iocchi et al. [1]. This feature is the most basic feature defined in the literature. In addition to this feature, as shown in Figure 1, the ability of robots to define their work, the coordination, and management style as a result of cooperation are the characteristic features of swarm robotics.

Another study of the swarm robotic definition was made by Erol Şahin. In this definition, besides the fact that collaboration is an indispensable option, robustness, flexibility, and scalability criteria [20] among the features in Table 1 were defined for the first time. These key features of swarm robotics can be explained as follows:

Robustness: One or more of the individuals in the swarm community may be disabled for various reasons such as malfunctioning, running out of battery, or being stuck on an obstacle. Robustness is the ability of the swarm to accomplish its task even in difficult conditions.

Flexibility: It is the ability to adapt and continue to work in changing conditions such as weather, environment, and task during the swarm mission. In flexible systems, changing conditions should be defined in the software of the robots and the robot should have a plan to continue to move using different parameters for different situations.

Scalability: The number of individuals in the swarm may be insufficient in some cases, and in such cases, new individuals may be added to the swarm from the surrounding swarms and additional swarm robot warehouses to complete the task. If the software and algorithms allow the number of individuals to be increased or decreased in a swarm, then the swarm is considered scalable.

Among the characteristics defined by Luca Iocchi et al., the control center in the swarm has a distributed structure rather than being centralized. This means that the robot can perform the given task on its own without taking orders from the central management. In the circumstances of encountering and failing to overcome the obstacle, other swarm robots in the neighborhood can interrupt their tasks to help the struggling robot. After the problem is solved with the aid of the other robots, the swarm robot continues to its main task. This feature is the known main characteristic of swarm robots, but it is not well developed in multi-robots. 

#### 2.2.1. Classification in Swarm Robots

In addition to the above characteristics, the physical characteristics of the swarm also reveal the structure of the system. Robot communities that do the same task cannot be called a swarm. In some cases, even missing a member in the group makes the mission fail. This type of robot community is often a multi-robot community. Another criterion for robot communities to be considered as swarm robotics is that the robots are of the same type, although not necessarily so. That is, the group should consist of the same robots as much as it is possible. Even if the group has a variety of robots, the group should not be too diverse for swarm robotics to work. In addition, different types of robots should not reduce the success of the task and should not interfere with the operation of other robots. Three different robot studies with hand-bot, foot-bot, and the drone can be given as an example in the Swarmoid project [21]. The task is achieved thanks to the working together of three different types of robots [20]. In swarm robotics, robots’ work is classified by the differences in their combinations. As the basis of swarm robotics is a technology inspired by nature, the term taxonomy is used instead of the term classification, as in biology. Here, some criteria are taken into consideration such as the tasks of the robots, the way they approach the problems, their communication, the way they are positioned, and the structures of the swarm. In this regard, the taxonomies proposed by Bayındır and Şahin, and Brambilla et al. can be prioritized. When both taxonomies are examined, the taxonomy criteria can be defined as follows:Number of individuals of swarm or size of the swarmCommunication ability of the swarm, perception capacity if any, and allowable communication distanceCommunication network methods such as addressing layer levels and data bandwidth established by the swarmProcesses of information processingVariability of the positioning of robotsRobot variety [22]

Accordingly, Bayındır and Şahin divide the swarm robots under the headings of problems, behavior design, communication, modeling, and analytical studies in their proposed taxonomy (Figure 2). In the taxonomy proposed by Brambilla et al., the swarm robots are gathered under the headings of methods and collective behavior (Figure 3).

#### 2.2.2. Classification in Swarm Robots

The idea behind the structural algorithms of swarm robots is imitating living things in nature. Similarly, their tasks can be considered identical with the behavior patterns of living things in nature. Accordingly, the tasks of swarm robotics can be obtained from the previously mentioned taxonomies in a way that mimics nature. Thus, the missions of swarm robotics can be classified as:ForagingCollectionTemplate formationTransportThe distribution of tasksCoordinated movementSelf—manufacturingMapping—DiscoveryTo decide

With the optimization methods mentioned above, the tasks to be performed by the swarm can be defined by being inspired by nature. These tasks appear in many areas of daily life. In some cases, more than one of these tasks is processed by the system at the same time. Naturally, having more than one task in swarm robotics depends on the power and support of the decision-making mechanism. We may briefly touch on some of these tasks. 

**Foraging:** It is a task based on finding the food that living things need in nature. It can be simulated by living things finding food and storing it at one point. An example of this is the collaboration of ants, birds, fish, and bees to find their food.

**Collection:** An example of this is the gathering of robots, like living things, at the desired point and completing the task by reinforcing the insufficient swarm to fulfill the task. The dispersal of the threatened swarm can also be thought of as a collection algorithm.

**Template formation:** This task is to ensure that individuals come together and form a shape. The movement of birds and fish by forming a certain shape, without hitting each other, is an example of this. This task can be simulated with the PSO method.

**Transport:** It is one of the collective behavior patterns frequently seen in some animal species. This behavior here can be seen as a combination of foraging and foraging behavior. An example of this is the transportation of an object by the swarm from one point to another, which would be too heavy for a single individual to carry easily. Providing the necessary workforce and transferring information within the group here forms the basis of this task.

**Task Allocation:** Performing different operations in complex tasks by dividing the swarm into groups will increase both the speed of execution of the operations and the success rate. While it is beneficial to divide the swarm into clusters, it is important whether the number of individuals in the swarm is sufficient or not. In addition, if there is a heterogeneous formation in the group, it is essential to check the suitability of different individuals for the task.

**Coordinated movement:** It is expected that the swarm, which consists of different structures, should cooperate, and complete the task in harmony. In a heterogeneous swarm, it may be necessary for another individual to complete the task that one individual in the team cannot do. Swarm robotics should be designed in a structure that will allow such formations.

**Self-manufacturing:** The main starting point of this task is the continuity of the swarm. Although the individuals in the swarm are numerous and dispensable, their losses can reach critical levels over time. In such cases, it may be necessary to reintroduce problematic robots to the swarm. This is an example of a self-manufacturing task. Apart from that, the swarm can work to create more complex parts from existing standard simple parts. This job is ultimately an assembly and repair job.

**Mapping-Discovery:** The task is to quickly identify an unknown region by scattering the swarms of robots and mapping the environment. Today, the idea of sending robots instead of people to very dangerous areas is accepted. As the area grows and the robot’s stay in the area gets shorter, it is more appropriate to have the swarm robotics do the job.

**To decide:** Distributed organizational structure is used in swarm robotics. At one stage of the task, while the whole group returns to the main task, a group of swarms can create additional tasks and move away from the main task. Or, if necessary, a part of the group may leave the swarm to remove an obstacle. In this task description, there is the decision-making process regarding all these tasks, such as coordinated movement, task allocation, foraging and transport in the swarm.

### 2.3. Communication Methods

If a single robot is unable to do the task, multiple robots must do it collaboratively and all robots in the swarm must work in harmony. This can be done through communication between the robots. In some cases, the swarm evaluates the situation and decides more individuals are required to complete the given task. To accomplish this, the swarm can communicate within itself and increase personnel for this task. The scalability feature from previously mentioned swarm robot characteristics is clearly needed for this [25]. Even if the number of members in the swarm changes, the defined task should continue without a new command being.

If there is a constant movement in the swarm, such as an increase or decrease in the number of members, then it is not expected that the swarm will have a stable communication structure. On the other hand, in the case where the number of personnel is fixed, communication cannot be established in the same way from one point to another on the same line because of the displacement of the robots in the swarm. Therefore, ad hoc communication topologies with a temporary dynamic structure are preferred in swarm robotics. In this sense, the activation or deactivation of the elements in the swarm does not disrupt the communication [22]. The communication methods to be established between robots are as follows:Mobil Ad Hoc Network—MANETSmart Phone Ad Hoc Network—SPANWireless Sensor Network—WSNInternet-based Mobile Ad Hoc Networks—iMANETVehicle Ad Hoc Network—VANET

As seen, network topologies used between robots are usually related to mobile networks. The reason for this is that the robots are not always at the same locations and the robots may be disabled. 

There are many hardware items and protocols to provide the communication methods that enable these topologies. It is important to know in which places the robots will work, when choosing the protocols and hardware structures based on these protocols. Since robots encounter too many obstacles indoors, transferring and receiving the signals consume more energy to pass the obstacles or due to the increase in communication distance. In such cases, even if the network structure is ad hoc, the energy consumed is problematic, and the signals may be interrupted. However, communication with less data is provided outdoors at a good rate, as there will be no obstructions. Since communication takes place over short distances in swarm robotics, this consumes less energy and enables precise communication. These communication methods are:Infrared communication (IR)Bluetooth communication (BTE)ZigBeeWi-Fi CommunicationLoRa Communication

These protocols and hardware structures together with the topologies enable the robots to communicate. However, one more point needs to be mentioned here. The studies generally describe the communication between individuals within the swarm or between the user and the swarm. However, there has been some discussion recently about the existence of more than one swarm. Naturally, considering that swarm robotics will become widespread over time, it is inevitable that there will be different swarms doing different tasks in the same environment. Moreover, these swarms may be heterogeneous swarms that do not have the same structure. In this case, it would be a great waste if the swarms do not work with each other. In cases where communication is not enough, it would be meaningless not to benefit from the individual of the other swarm found on the neighborhood. In this context, recent designs in accordance with Figure 4 have been made by adding algorithms in which different swarm groups can work together. With a study on this subject, Ali et al. [26] created an idea in this direction and researched the idea that communication would be as follows:User-to-swarm communicationIntra-swarm communicationInter-swarm communication

As a result of this study, it was seen that the most highlighted feature was that communication continued in different swarms, and communication within the swarm was completed by the other swarm. As it can be seen in Figure 5, 46% of the communication load was realized in an inter-swarm way.

Network communication is a very parametric concept as agreed by the field researchers. As a result, the communication can be customized for effectiveness by adjusting these parameters. This is very critical in most communication designs. In some cases, it is not desired that the nodes belonging to other networks can participate in communication with the nodes forming the network; in contrast sometimes, as in the previous example, it is desired that the communication is inter-network and global, with close communication with the external environments. Providing these two different requirements is ensured by communication parameters and can be achieved via hardware or software systems. The parameters that must be defined during communication are mentioned below. 

Communication range: Communication takes place between individuals of the swarm and it is obvious that the distance of communication between them is important. While the distance between two robots is a few units, increasing this distance to a few hundred units creates unnecessary costs and redundant interference. Therefore, the distance between the robots for effective communication defines the communication range. An excessively large range increases the number of communication operations needlessly and also causes individuals to be unable to communicate.

Communication area: Under normal conditions, the robot makes a circular communication in such a way that there is no empty angle around it, or we assume that it does so. However, this circular structure may be disrupted due to reasons such as obstacles around the robot and the characteristic features of the antenna used. In such a case, the robot cannot exhibit the expected behavior as it will stop giving and receiving signals. Therefore, it cannot send data to the next robot and may not receive data itself. The behavior of the swarm may deteriorate, and the mission may be compromised.

Length of messages: During communication, the message does not consist of only the data content. In addition to the data content, a payload is needed consisting of a header formed by parts such as destination address, source address, message size, CRC control, etc. These additive sections are to ensure that the message arrives correctly. Sometimes the payload part may be as large as the data part. Therefore, the size of the header increases the length of the message.

Message propagation time: In some cases, when the message is sent it may be requested to actively roam in the swarm for a while to guarantee to reach all individuals. Keeping the message duration time long will cause the message to be repeated in the swarm; therefore, there should be a determined limit value to stop spreading the message from individual to individual.

Interactions: The most important criterion of cooperation in swarm robotics is the communication between them. Communication blockages, interruptions and interference with other unrelated information makes the interaction between the robots difficult. In addition, the task to be done may fail if individuals cannot get the correct data. Communication should be designed to eliminate these problems [27].

### 2.4. Optimizing in Swarm Robots

It is not possible to control each of the robots in the swarm one by one due to a large population of robots. Robots must move alone to complete the task they are taking. Thus, robots must overcome the difficulties encountered on their own and solve the problems that may arise on their own. The individual who overcomes these difficulties should adjust its movement according to the swarm. 

This kind of movement of the swarm takes place thanks to the algorithms to be written. The starting point when defining these algorithms is nature. There are many creatures in nature that behave similarly in this way. These creatures mostly move together to find food and escape from their predators. During this movement, many creatures in the swarm reach their destination in swarms, often without colliding with each other and very quickly. They also respond just as rapidly to a change in target. Researchers have defined nature-inspired meta-heuristic optimization methods based on this harmony in the behavior of animals in nature. These optimization methods are designed to describe the control of individuals in the swarm by simulating nature and are very diverse [28]. Some of them are:Particle Swarm Optimization (PSO)Ant Colony Optimization (ACO)Artificial Bee Colony Optimization (ABC)Fish School Search OptimizationCuckoo Search OptimizationFirefly AlgorithmBat AlgorithmFlower Pollination AlgorithmGray Wolf OptimizationElephant Swarming OptimizationCrow Search OptimizationRaven Roasting Optimization Algorithm

Most of these optimization methods are used in swarm robotics studies. Here, we will focus on the most effective algorithms in swarm robotics.

**Particle Swarm Optimization (PSO):** This optimization is a collective action algorithm inspired by the behavior of animals that move collectively, such as swarms of fish and birds, to find food and avoid predators. It was first described by James Kennedy and Russell Eberhart in 1995 [29].

The behavior of a swarm that collects for foraging is studied in Particle Swarm Optimization. This behavior of the swarm is formed as a vectorial combination of the behavior of the individuals arising from the experience and the states of the individuals initially. To achieve the target in the most efficient manner, the swarm determines its and the swarm’s best values and attempts to position the swarm according to that point based on the swarm’s current best value.

The mathematical model of the vectors in the graph of Figure 6 is as proposed by James Kennedy and Russell Eberhart in 1995.
(1)$$v_{i}\left(t+1\right)=wv_{i}\left(t\right)+c_{1}*r_{1}*\left(pbest_{i}-x_{i}\;\left(t\right)\right)+c_{2}*r_{2}*\left(gbest_{i}-x_{i}\;\left(t\right)\right),$$
(2)$$x_{i}\left(t+1\right)=x_{i}\left(t\right)+v_{i}\left(t+1\right)$$

**Ant Colony Optimization (ACO):** Ant Colony Optimization is a path-finding algorithm for ant colonies first proposed by Dorigo et al. [30]. In their work, they called their system the ant system. They named the model suitable for this system as the Ant algorithm. The ant, when foraging, leaves an odor called a pheromone at the points it passes while on the move. At first, the smell is fresh. Every passing ant leaves a pheromone on this spot. Thus, since the amount of pheromone will be higher on the shortest path, the choice will be on this path. Which path the ant will choose when it comes to the intersection is determined by the randomness value. This random situation creates an opportunity to find better ways. There may be free-moving ants outside of this path, but since the pheromone they will leave will be less and will disappear over time, ants naturally prefer the shortest, that is, the most-used path [31].
(3)$$\tau_{xy}\leftarrow \left(1-\rho \right)\tau_{xy}+\sum_{k}^{m}\Delta \tau_{xy}^{k},$$
(4)$$\Delta \tau_{xy}^{k}=\{\begin{array}{c} Q/L_{k}\\ 0 \end{array}\;other\;case\;if\;the\;ant\;uses\;the\;xy\;curve\;in\;k\;rounds$$

**Artificial Bee Colony Optimization (ABC)**: Bee Colony Optimization is an optimization algorithm based on feeding. In this optimization, it is one of the characteristics of swarm intelligence that bees can distribute tasks without central management and organize themselves [32]. To find nectar, bees take long routes that can be in different directions. In this optimization, the first step is for the bees to come to the hive to share the information and direct the other bees to the nectar [33]. The other step in this optimization relates to the weakening of the source, and the bees’ switching to different nectar searches. In this case, the worker bees that come for nectar join the scout bee swarm to find a new nectar area. The scenario is processed step by step to explain the working of the algorithm [34].

The bee will find the food source around the hive. The algorithm starts by generating random points between (0,1) within this area:

*x_ij_*, *i* = 1,…, *N*, *j* = 1…*M*, where *N* is the number of food sources and *M* is the number of parameters to be optimized. *x_min_* is the lower limit of the *j* parameters.
(5)$$x_{ij}=x_{j}^{min}+rand\left(0,1\right)\left(x_{j}^{maks}-x_{j}^{min}\right)$$

One of the constraints in this algorithm is that each resource has one attendant bee. This leads to the assumption that the number of food sources is equal to the number of employed bees. The worker bee wanders around the sources, and if the source they find is better than the previous one, the other source is forgotten and the new source is memorized. Here *φ* is again a randomly generated integer. *v_i_* is the source in the *x_i_* neighborhood. *x_k_* is the neighboring solution. As the difference between *x_ij_* and *x_kj_* decreases, the optimal solution is approached.
(6)$$v_{ij}=x_{ij}+\varphi_{ij}\left(x_{ij}-x_{kj}\right)$$

In case the *v_ij_* value produced in this process exceeds the lower and upper limits previously specified, Equation (7) is shifted to the lower or upper limit values of the *j* parameter.
(7)$$v=\{\begin{array}{cc} x_{j}^{min} & v_{ij}<x_{j}^{min}\\ v_{ij} & x_{j}^{min}\leq v_{ij}\leq \\ x_{j}^{maks} & v_{ij}>x_{j}^{maks} \end{array}x_{j}^{maks}$$

The cost value of this resource is *f*(*v_i_*). Here, the *v_i_* value produced within the limits is substituted in Equation (8) and the fitness value of this solution is calculated. The bee chooses between *v_i_* and *x_i_*. If the newly found value gives a better result, the old information is deleted and a new one is taken instead. *error_i_* is reset. If the new resource is not better, the *error_i_* is increased by 1, and the bee continues to search for the resource.
(8)$$fitness=\{\begin{array}{cc} 1/\left(1+f_{i}\right) & f_{i}\geq 0\\ 1+abs\left(f_{i}\right) & f_{i}<0 \end{array}$$

$f_{i}$, is the cost value of the $\vec{v_{1}}$ resource.

After completing their research, the bees return to the hive and convey their knowledge to the onlooker bees by dancing. Criteria such as nectar rate and distance of onlooker bees are represented by the fitness value. Probability values can be calculated with Equation (9) based on the fitness value that the onlooker bees will use while making the selection process.
(9)$$P_{i}=\frac{fitness_{i}}{\sum_{j=1}^{N}fitness_{j}}$$

In the selection process according to the roulette wheel, if the φ value produced in the range of [0, 1] for each resource is greater than a randomly generated value, the onlooker bees use Equation (6) to produce a new resource in this resource region and apply the selection process between *v_i_* and *x_i_*, so the better one is selected. If the *x_i_* solution has not improved, the solution is preserved and the *error_i_* is increased by 1; if it is advanced, the *error_i_* reset is performed. This step is repeated until all onlooker bees have dispersed to the food source areas.

As the onlooker and attendant bees go to the source, the incoming nectar information is checked. If this value falls below the threshold amount, the resource is considered to be exhausted. In this case, the employed bees turn into scout bees. After this moment, they start searching for new solutions using Equation (5) [35].

**Firefly Algorithm:** Fireflies approach each other by a brightness criterion, regardless of gender. Since the distant firefly will be less bright, its attraction will be less. In addition, the less bright firefly will move towards the bright ones in the environment. If there are no fireflies around, the firefly will move randomly. The basis of the optimization method is based on this movement mechanism of the firefly. It is modeled and formulated by Xin-She Yang [36].

**Bat Optimization**: It is an optimization method inspired by the bat’s echolocation behavior. Bats create a kind of radar signal by using sounds at frequencies inaudible to the human ear to locate their prey and determine their direction. Thanks to the echo of this signal, they can communicate with each other and perform their survival activities. They can achieve their goals with values such as the frequency of the signal they send and the return time. It is modeled and formulated by Xin-She Yang [37].

**Cuckoo algorithm:** (Incubation Parasitism) It is an algorithm originated from the migration and breeding strategy of the cuckoo. The cuckoo waits for the owner of the nest to leave and leaves its young in a foreign bird’s nest. The cuckoo, who comes to the nest after the owner has left, may throw previous eggs from in the nest and places its own egg in the nest. Placement is important because if the nest owners detect fake eggs when they return to their nests, they will throw the egg out of the nest. If the eggs provide credibility, the real nest owner will adopt the eggs. The cuckoo’s eggs usually hatch earlier than the nest owner’s eggs. Early hatching cuckoos throw other eggs from the nest, so that the nest owner takes care of them only. If the opposite occurs, and they come out later the cuckoo cubs, which are greedier and more aggressive in terms of nutrition, starve the nest owners’ offspring, preventing them from sheltering in the nest. Even if the owners of the nest are aware of the situation, they cannot harm the cubs because they adopt the cuckoos with their parental instinct. In this way, the adult cuckoo enters the migration period early and leaves the care of its young to another bird. This model was developed in 2009 by Xin-She Yang and Suash Deb [38].

### 2.5. Optimizing in Swarm Robots: Real Life Examples

Because of its inspiration from biological life, swarm robot optimization produces work that can be applied in real life. Several examplar studies will be mentioned to emphasize the importance of swarm optimization methods. In one study by Li et al., a new approach combines Wireless Sensor Networks (WSN) and Multi Mobile Robots (MMR) topics and discusses cooperation between them to obtain swarm intelligence [39]. In that swarm intelligence application, three mobile robots and seven WSN nodes are used for simplicity. The robots nearby are detected by the static WSN nodes, and a virtual entity assigned. The attributes, stored in WSN nodes, of the created entity are shared with the nearby robots because of the limited communication range. This approach eases the remote control of robots and their formations in the swarm without making any change of the hardware and software of the robots. Another study is about dynamic cooperation of multirobot systems maintaining a pre-defined swarm formation in a trajectory following by Lee et al. [14]. The movements of three CRX 10 mobile robot platforms are observed with externally positioned cameras in an indoor area. Robot formations are controlled by using camera data and individual robots solve their local optimization problems on their own. Experimentally tracking a circular trajectory in triangular and line formation of the swarm is achieved by using a dynamic cooperatively co-evolving PSO algorithm. In another application, drones carrying RFID readers check inventory of raw materials and finished products dynamically in a factory environment studied by Liu et al. [18]. Considering that drones as aerial robots have their flight time limitations, their effective trajectory routing with optimization is critical. Here the proposed method successfully used drone swarms for accurate and fast inventory checking. Another application is about autonomous aquatic surface robots controlled by self-organized evolutionary techniques, developed by Costa et al. [40]. The robots here are manufactured by CNC milling and 3D printing and have an open-source software available to be developed by other researchers. Robots are equipped with multiple sensors such as GPS navigation, digital compass, and temperature sensors. Robots move in swarm mode and their task is organized using evolutionary optimization algorithms.

## 3. Results

### 3.1. Design Problems in a Swarm Robotics

Moving a single robot is usually easier than moving multiple robots. In this study, research has been done on the problems that can be experienced when it is needed to move the swarm together, and optimization suggestions are presented according to these problems [10].

#### 3.1.1. Problems Caused by Optimization Design

**The area occupied by individuals in space**: Calculations made in optimization design usually do not calculate the volume of the individual and see it as a point. However, as seen in Figure 7, an individual has a mass and a volume that occupies space. In the optimization design of robots, this detail must be added to the equation. However, the robots may not be of the same type. As a result, different covering areas will be created for each robot. Generally, it may be desirable for the robots to avoid contact and maintain a certain distance from other robots depending on the task.

For the solution of this design problem, the physical conditions of the robots should be considered when calculating the coordinates of the individual, both in the equation and in the algorithm calculations. Overlapping conflicts that may occur should be eliminated by software arrangements. For the case of different types of robots, this value should be calculated separately for each different group. In the case where contact between robots is not desired, taking the bat optimization as an example and using the echo system, the proximity distance to the other robot should be limited by sensors such as infrared and ultrasonic (Algorithm 1).
**Algorithm 1** Solution: The area occupied by individuals in space1:**if** (robot radius + tolerance) < distance to target **then**2:  Keep moving3:**else**4:  Stop motion5:**end if**

**Conflict of Coordinates:** If robots are controlled from a single place in the algorithm, there is a possibility that two robots can be in the same coordinate at the same time. This is a problem as can be seen in Figure 8. The fact that the robots are headed towards the same target point reveals the possibility that there will be contact with each other, and serious collisions may occur. This is possible in experimental simulation environments, but in the physical world, this is not possible. Robots that want to physically go to the same point can also make undesirable movements such as climbing on top of each other, apart from bumping into each other with structural compatibility.

In the case of central control in algorithm equations, overlapping in coordinates can be eliminated by software control of coordinates, and sending overlapping coordinates to individuals can be prevented. This software elimination may cause an increase in the number of iterations, but delay time sourced by sending incorrect coordinates to the robots will be reduced. As another method of preventing coordinate conflicts, distance control is provided by the echo method. Even if incorrect coordinates occur, priority is given to the sensor data in the algorithm of the robots, so that the problems of incorrect codes will be prevented (Algorithms 2 and 3).
**Algorithm 2** Solution 1: Conflict of Coordinates1:**for each** Robot **do**2:  **for each** other robots **do**3:    **if** robot coordinates are not equal to other robots coordinates **then**4:      Accept the coordinate5:    **else**
6:      Choose different coordinate7:    **end if**
8:  **end for** other robots9:**end for** robots

**Algorithm 3** Solution 2: Conflict of Coordinates1:**if** Desired range > Measured range **then**2:  Keep moving3:
**else**
4:  Stop motion5:
**end if**


**When the target is given to the robots**, all the robots will naturally move towards this point. But that does not make any sense, because there is only one point, as in Figure 9. All robots cannot reach this point. Therefore, there will be continuous movement in the robots. The continuous movement that occurs will cause both the processor and the communication to be busy during the task. Battery problems will occur because of continuous motion energy, processor, and communication tasks.

The solution to this can be solved by distance control and the approach of robots to the target point. The movement of the robot approaching the target point can be stopped by examining its approach to the previous robot and stopping the motion at the required approach point. Stopping can be done with the equalization system or by creating communication calculations (Algorithm 4).
**Algorithm 4** Solution: When the target is given to the robots1:**if** (robot radius + tolerance) < distance to target and desired range > measured range **then**2:  Keep moving3:**else**4:  Stop motion5:**end if**

**Obstacles in the working area:** At the working point, as can be seen in Figure 10, there may be obstacles that may prevent reaching the target on the target path of the swarm. Obstacles may not allow or limit the passage of the swarm at one point. In addition, there may be routes where these obstacles are absent or less. The swarm may do many iterations to enter and exit a path closed by obstacles. As a result of these attempts, their energies may reach points where they cannot fulfill the task.

The software that will send the swarm to the target may then use more than one optimization effort. By using Bee Colony Optimization, the path to the target can be determined by sending scout robots that can find the shortest path. With the detection of the target, the scout robot, which gives communication information to the swarm, can direct the swarm to this path. If there is more than one robot crossing the obstacle, the shortest path can be calculated, and the swarm can pass through this path. Using Ant Colony Optimization, the swarm is divided into several groups. The shortest path from the robot groups to the target from several different paths can be shared by the swarm, and the swarm can be directed to this path over time. In the meantime, individuals whose batteries are below a certain level, in case the distance between the swarm and the target is large, provide the relay point that will provide communication instead of reaching the target point, as in the cuckoo algorithm, thus providing the maximum benefit to obtain from before their batteries are completely exhausted. If the swarm reaches the target and there are no robots left behind, these robots can perform their movements to the extent of their battery charge. In cases where their batteries are not enough, they can share their locations and ensure that they are left at the charging stations at the end of the task or get support from other robots when necessary (Algorithm 5).
**Algorithm 5** Solution: Obstacles in the working area1:**for each** robot **do**2:  **if** robot arrived at the target **then**3:    add robot arrival time to list 4:  **end if**
5:path = shortest path in the list6:robot = the robot with the shortest path7:**end for**

**Determination of the gathering point of the swarm:** While living creatures in nature move in swarms, their behavior patterns are determined by finding food and avoiding predators. These movements determine their goals. In the robotic swarm, it is necessary to define the target. While the swarm whose initial movement is given by the user is moving towards the target, a change of the target’s position should also change the movement in the swarm. As in Figure 11, the swarm must determine the working area without the knowledge of the user, find its target autonomously, and head towards the target.

In nature, the food point is never defined. The swarm is focused on food rather than coordinates. Similarly, it is not clear where the hunter will come from. Therefore, instead of the location of the target, the characteristics of the target should be defined, and the target should be monitored continuously. Here, faulty spots can be distinguished from targets by using object recognition systems or distinctive signals or labels on the target.

#### 3.1.2. Problems Caused by Communication

Communication between robots is very important in swarm robotics. Moving the robots together, reaching the target, and completing the task is a process undertaken together by the swarm. During these processes, cooperation can only be realized through communication. As a result of the disappearance, interruption, or deterioration of communication within the swarm, the swarm cannot move properly.

**Exceeding the communication distance:** Communication between robots takes place in the form of receiving and then relaying these data. Communication between points is in an ad hoc manner. If the physical distance between the moving robots exceeds the transmission distance, as in Figure 12, the communication link is broken. In case of disconnection, more than one of the robots may be disabled.

For individuals to communicate with each other, the communication signal must reach to a certain distance. The communication distance of some robots must be designed as greater than all other robots in the swarm to prevent any robot from being disabled because of the physical distance between the robots. This design increases the coverage area and may prevent disconnections. Therefore, some robots in the swarm may have higher transmission power and receiver sensitivity to increase the communication distance. The presence of such robots, though few in number, can increase the communication distance and ensure uninterrupted communication throughout the whole swarm (Algorithm 6).
**Algorithm 6** Solution: Exceeding the communication distance1:**if** incoming message data available > 0 **then**2:  incoming message is used3:**else**4:  main message is used5:**end if**

**Communication distance is too high:** If the communication distance is increased too much, the transmission circuits are busy for a longer time and the battery is consumed faster due to the increase in the transmission distance. In addition, the probability of interference of signals increases. As a result, this can disrupt their tasks and make their battery run out faster.

Short communication distances of the swarm robots cause the communication chain to break while keeping it long causes echoes, as in Figure 13. The solution is to form the communication in a multi-hop manner. For this, as in the previous problem, a communication channel can be used for the long distances and a separate communication channel for the short distances (Algorithm 7).
**Algorithm 7** Solution: Communication distance is too high1:**if** incoming message data available > 0 **then**2:  incoming message is used3:**else**4:  main message is used5:**end if**6:**if** the message is new **then**7:  use the message8:**else**9:  don’t use the message10:**end if**

**Change in the target point:** The same problem experienced in optimization designs can also be experienced in communication points. As seen in Figure 14, the target may be mobile around the swarm, or there may be more than one. Directing the whole swarm to the right destination is a decision that must be taken within the swarm. This is the key characteristic of swarm robotics. This decision should be transferred to the whole swarm at the same time.

This event needs to be conveyed very quickly. Otherwise, some individuals in the swarm may move towards opposite points from the swarm.

The simplest way for the swarm to take a decision very quickly and to transfer it to the whole swarm in the fastest way will be the continuation of the solutions given in the previous problems. Information will be shared quickly by individuals as a result of the transmission of information to all individuals by broadcasting through different channels of communication in the swarm and not reprocessing the message after acknowledgement. Individuals or nodes that receive the information quickly transfer it to the closest individual. In this way, the entire swarm can be directed to the target at the same time.

**Communication confusion between swarms in tasks consisting of different swarms**: As known from swarm robotics studies, there may be swarms with different tasks in the same environment (Figure 15). Basically, the different communication parameters of these swarms will prevent communication with each other. However, as we have seen from previous studies, this may not be disadvantageous as expected. Therefore, communication between the swarms may be a beneficial situation in terms of task completion. However, if two swarms use the same communication channel, it will cause communication confusion.

To overcome this problem, groups working in different channels can communicate using common channels when necessary. In this way, when necessary, the interrupt message to be sent to the robots focusing on their own work is provided to communicate with the other swarm in the same area, and the swarms can help each other to increase the success of the task. Thus, this action can create a larger swarm. Normally, in the smaller community, the group using low-overhead communication will move faster as it will use a smaller communication volume to process given transactions. As a result, communicating with the other swarm in the same area provides the necessary communication and increases the success of the mission with the help of other swarms. If the communication channels are different, the swarms will not affect each other.

#### 3.1.3. Swarm Robotics Study Using PSO

The system to be mentioned here was designed using the Webots [41] simulator, with 2 different robot swarms and 10 robots in each swarm, with a total of 20 robots, as seen in Figure 16. Each swarm will communicate internally in one channel, and when necessary, the next upper channel will be used to take the two swarms to a single point [42]. The robots to be used in the swarm in the simulation software are Elisa-3 robots created by GCtronic [43]. These robots are equipped with GPS Compass sensors on them [44].

A swarm was controlled by the PSO method, which would answer most of the problems identified in this study. Basically, here the intention is to bring the swarm closer to a point based on the PSO algorithm. However, while the control is calculated at one point based on PSO, this contradicts the distributed architecture of swarm robotics. Hence, the PSO algorithm is run separately on all individuals in the swarm instead of a single point, and the pbest value is calculated by using a mathematical equation. The pbest values for individuals will be different due to the different positions of each individual and the random r_1_ and r_2_ values in Equation (1). These calculated coordinates are transferred to all individuals in the swarm. Individuals in the swarm determine the gbest value for the swarm by processing the calculated pbest data for each incoming individual. In the mathematical function, the gbest value among the coordinates of all robots will be considered the same by all individuals in the swarm. Consequently, there is one optimized value for the whole swarm in reaching the target.

In Section 3.1.3, the modeling of the PSO is provided with control software in a distributed architecture suitable for the characteristics of swarm robots, where individuals will be compatible with the group but independent within the group [1]. In the optimization, a study has also been made to include the problems mentioned in Section 3.1.1 and Section 3.1.2. Thus, coordinate conflicts, unnecessary behaviors in the control of the swarm, interruption of communication or signal echoes are eliminated, and solutions are applied in swarm robotics that move using PSO algorithm. In Algorithm 8, this algorithm structure is given in the pseudocode. Although there are simulations convenient for many scenarios in the application, there are still situations where real-life conditions cannot be met. For example, the attenuation of the signal strength due to the battery conditions of the robots and obstacles are not included in the equation. It is appropriate for the development of the project to consider this in future studies.
**Algorithm 8** General Solutions1:**Initialization:**2:**for each** of the dimension **do**3: Set random initial values for particle (position and velocity)4:**end for**5:**while** (radius of robot + tolerance) < distance from target point **do**6: **for each** of the particle—1 **do**7:  **if** there is communication **then**8:   receive particles information in swarm (position, velocity and particle)9:  **Else**
10:   increase the communication range11:   receive basic swarm data (position, velocity and particle)12:  **end if**13:  **if** the message is new **then**14:   use Message15:  **end if**
16: **end for**17: collect information of all robots (position, velocity)18: **for each** of the particles and dimension **do**19:  for each personnel evaluate the fitness: f(position)20:   **if** personnel best > global best **then**21:    global best = personnel best22:   **end if**23:  update information of all robots (position, velocity)24:  **end for**
25: **end for**
26: send robot information position, velocity27: **if** approach distance > distance sensor value **then**28:  keep moving29: **else**
30:  stop motion31: **end if**
32:**end while**

As seen in Algorithm 8, for this application the pseudocode of the software is given to solve the problems arising from the optimization design mentioned in the study. Accordingly, the collisions of robots with each other and overlapping coordinates are prevented using this algorithm. In addition, the swarms communicated independently through different channels for the solution of communication-related problems. For achieving this, the communication links are designed to allow interoperability when necessary. Thus, the robot, whose communication is interrupted from the swarm, is enabled to act as a part of the other swarm. In this way, the robot, which could not communicate in the field, is also prevented from being disabled.

## 4. Discussion and Conclusions

In this research, the nature-inspired optimization methods used for swarm controls and the communication methods within and between the swarms are emphasized. Thus, the problems encountered in swarm robotic behaviors can be overcome using these above-mentioned problem-solving methods found among living creatures in nature.

In this study, the optimization and communication design problems that can be experienced during the realization of the movements of swarm robotics are stated. Using several different algorithms instead of using only one for swarm control yields a better solution to the problems emerging in swarm robotic design. Some methods were suggested suitable for swarm robotics architecture using different approaches to some classical optimization methods such as PSO.

It is concluded that the use of heterogeneous communities which are also based on swarm robotics can clearly increase the swarm’s success. In nature, there are many examples of this phenomenon. The presence of individuals with different duties in the same environment in bee and ant colonies is a good case. In addition, the swarm can be formed from individuals who have different tasks with the same purpose, in order to have a robust structure, better control, and fast communication.

Although the homogenous employment of individuals is essential and a designed coordinator is less often suggested in the distributed control architecture used in swarm robotics, some individuals designed with different purpose from the same swarm can be selected to act as coordinators to improve this common situation. When necessary, this individual’s duty can be exchanged with that of another individual. This approach will help to eliminate the requirement of creating individuals with different architectures and remove the need to search for the replacement of an individual that may be disabled for some reason in a group. The swarm designed in thimanner will be a great team that makes decisions on its own, provides high cooperation within itself, has strong coordination, does not have problems in adding or subtracting individuals, and adapts quickly to task changes. When required, the idea of dividing the swarm into parts and combining them for a single task will speed up the inter-communication and will also be more effective in accomplishing the task by sending or recruiting a team from within the swarm to the other swarm.

In future works, it is planned to interpret and optimize the simulation data with deep learning and optimization algorithms. Afterwards, it is planned to create a swarm robot application in a virtual environment to perform the pre-defined tasks. The results and experience obtained from this will be transferred to the designs in the real environment. Smart agriculture applications are good candidate for the next stage of this study.

## Figures and Tables

**Figure 1 sensors-22-04437-f001:**
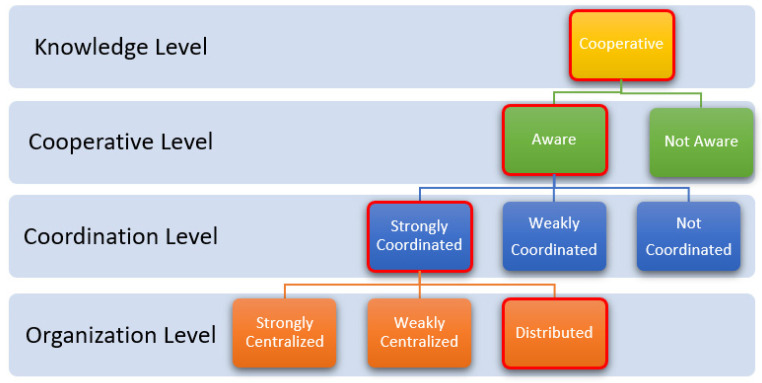
Characteristics of swarm robots [1].

**Figure 2 sensors-22-04437-f002:**
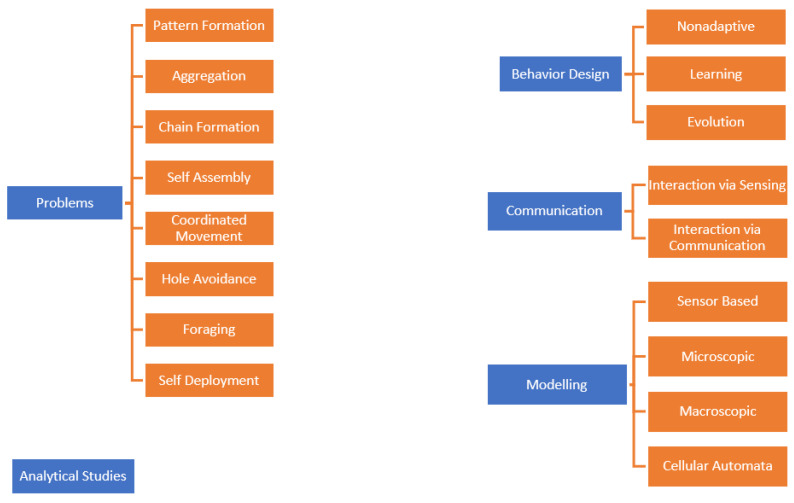
Taxonomy proposed by Bayındır and Şahin [23].

**Figure 3 sensors-22-04437-f003:**
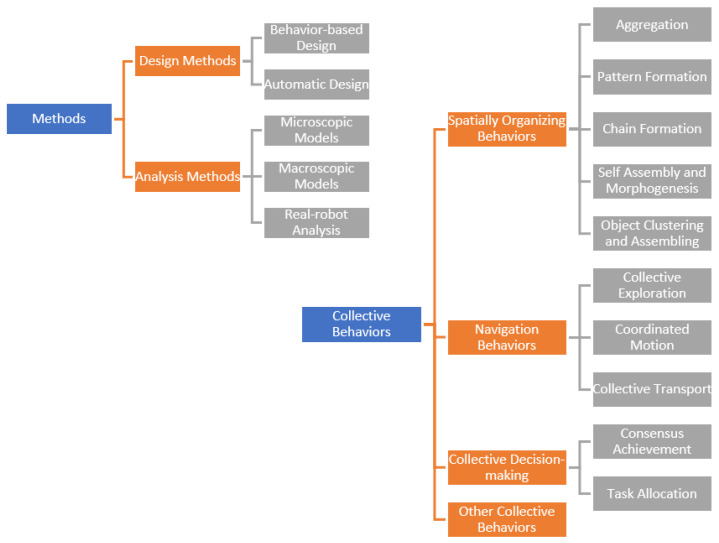
Taxonomy proposed by Brambilla et al. [24].

**Figure 4 sensors-22-04437-f004:**
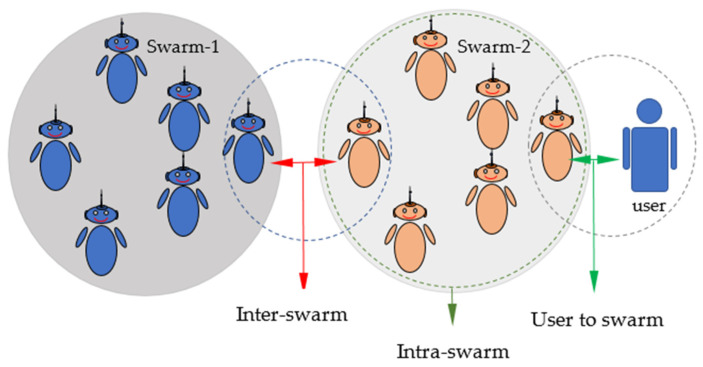
Communication models used in swarm robotics [26].

**Figure 5 sensors-22-04437-f005:**
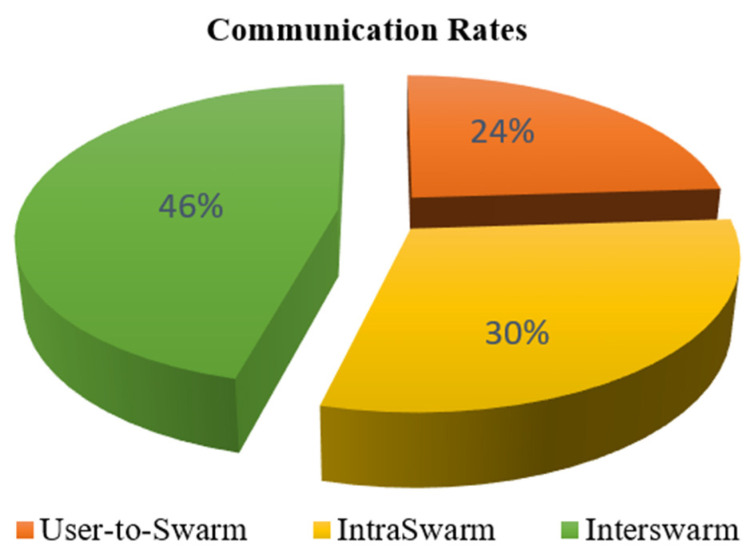
The influence of various methods of communication on the network [26].

**Figure 6 sensors-22-04437-f006:**
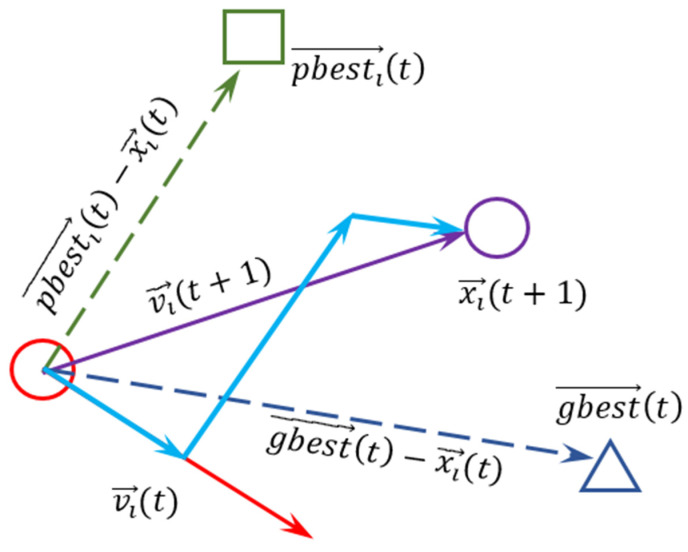
Behavior of the individual in Particle Swarm Optimization.

**Figure 7 sensors-22-04437-f007:**
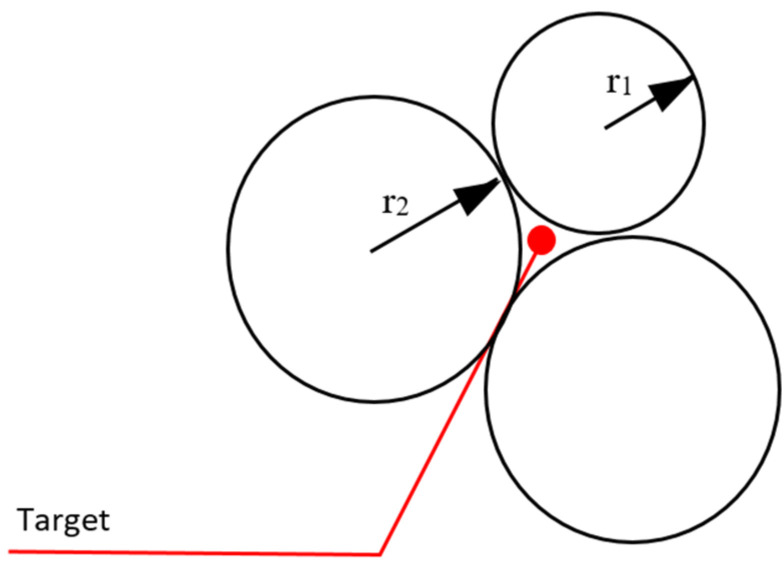
Approach of robots to target point.

**Figure 8 sensors-22-04437-f008:**
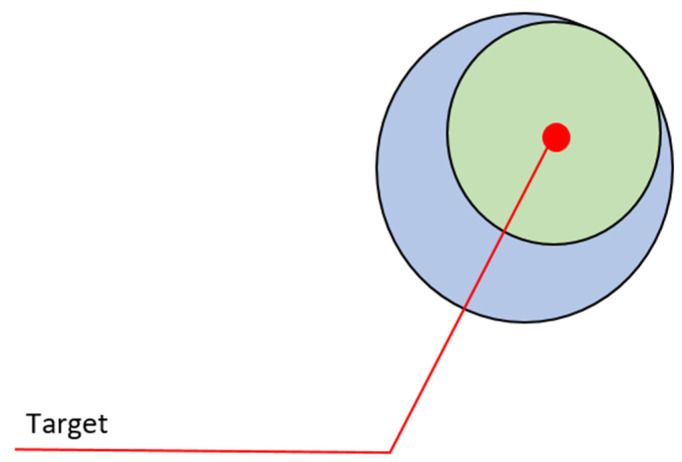
Defining the same coordinate to different robots.

**Figure 9 sensors-22-04437-f009:**
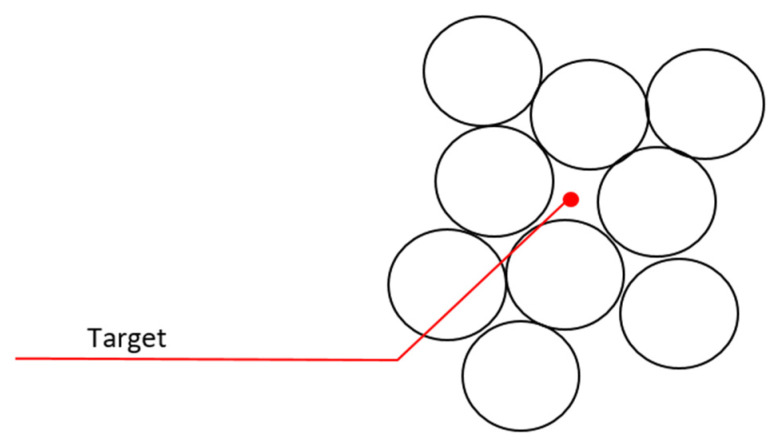
Inability of the individuals in the swarm to reach the target.

**Figure 10 sensors-22-04437-f010:**
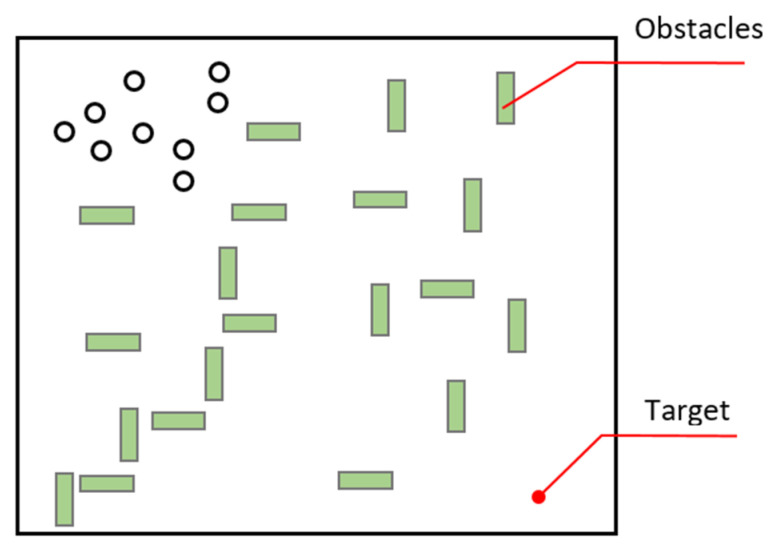
Obstacles in the work area.

**Figure 11 sensors-22-04437-f011:**
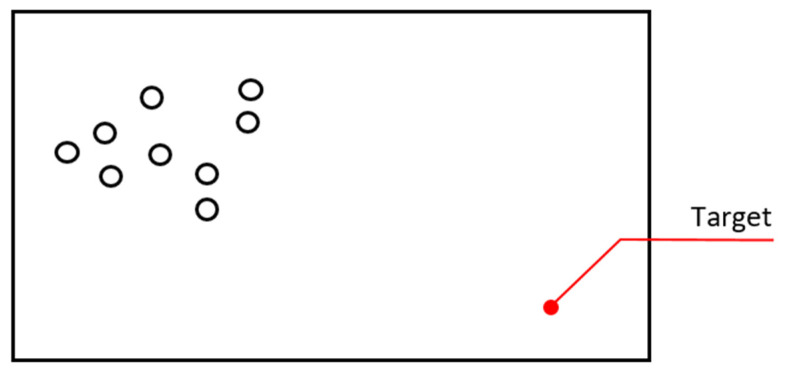
Determining the working area, determining the target.

**Figure 12 sensors-22-04437-f012:**
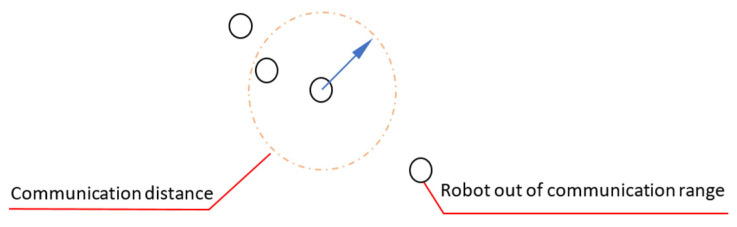
Robot out of communication range of the swarm.

**Figure 13 sensors-22-04437-f013:**
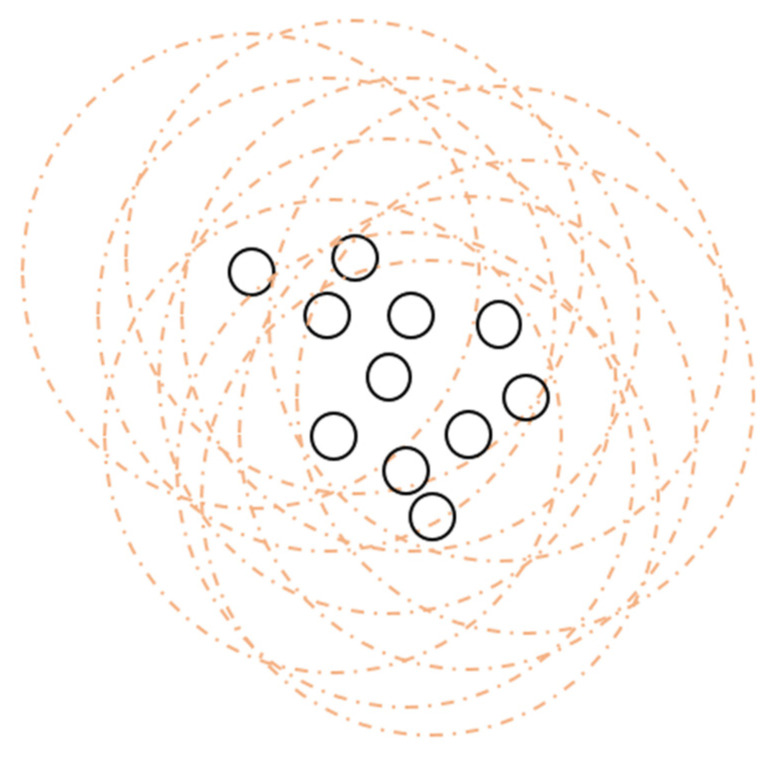
Keeping communication ranges long.

**Figure 14 sensors-22-04437-f014:**
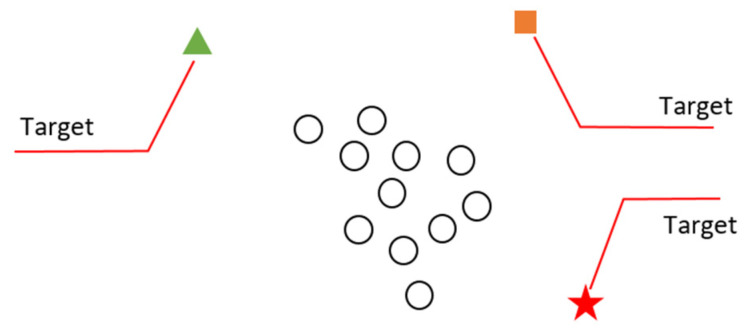
Communication status of swarms with different missions in the same area.

**Figure 15 sensors-22-04437-f015:**
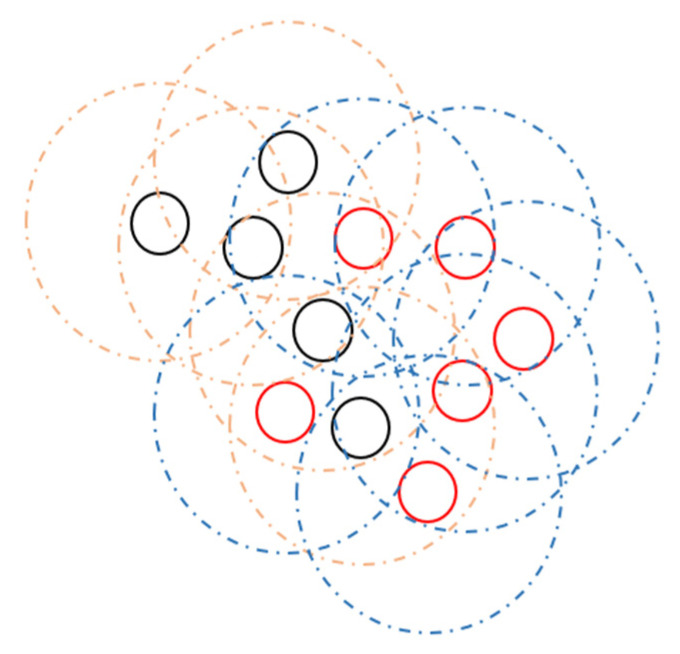
Different goals in the workspace.

**Figure 16 sensors-22-04437-f016:**
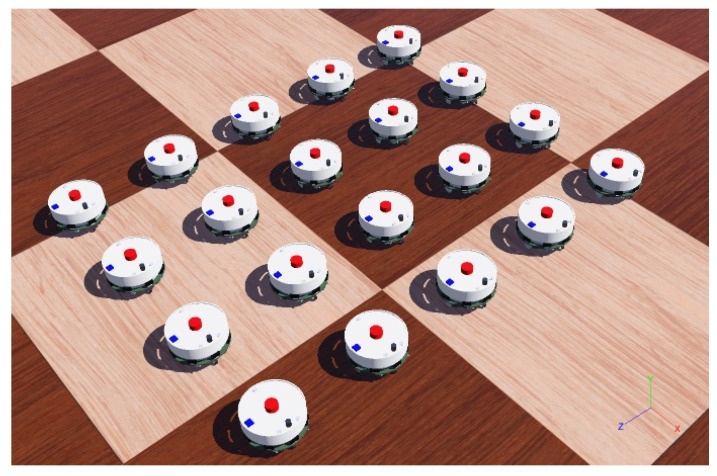
Elisa-3 robot swarm in Webots simulator.

**Table 1 sensors-22-04437-t001:** Distinctive features of Multi Robots Systems [19].

Features	Swarm Robotics	Multi Robots	Multi Agents	Distributed/Parallel Computing
**Number of elements**	Dozens—Thousands	Dozens—Hundreds	Dozens	Dozens—Hundreds
**Control type**	Central/Diffuse	Central/Diffuse	Central	Central
**Element type**	Same	Same/Different	Different	Same/Different
**Flexibility**	High	Low	Middle	Low
**Robustness**	High	Middle	Middle	Low
**Scalability**	High	Low	Middle	Low
**Environment**	Unknown	Known	Known	Known
**Usage areas**	Where precise mission success needed	Multiple robot applications	Source control and monitoring tasks	Calculations that require a lot of math

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
