# Peer review of "Usage of Evolutionary Algorithms in Swarm Robotics and Design Problems"

_sensors, 2022, doi:10.3390/s22124437_

Round 1
Reviewer 1 Report
In this paper, the authors consider the general structure of swarm robotics. Algorithms inspired by nature, which form the basis of swarm robotics, are introduced. The topic is quite interesting and this paper is well-written. I have the following comments for the further improvement of the paper.
1) The authors are suggested to add one more keyword in the paper.
2) The references should be cited in the Introduction part, and also give some descriptions of them.
3) Please fix some figures with minor typoes such as Fig. 2 and Fig. 3.
4) The authors are suggested to present a pseudocode or procedure for the proposed algorithm in the paper.
5) The reference part is suggested to be updated with some other applications on evolutionary algorithms to enhance related and previous work part, e.g., Three-dimensional route planning based on the
beetle swarm optimization algorithm. A novel convolutional neural network model based on beetle antennae search optimization algorithm for computerized tomography diagnosis.
The paper is good writing and presents technical contributions, which could be accepted after a revision.
Author Response
Thank you very much for your valuable suggestions. We have addressed them during the revision of the manuscript.
- The authors are suggested to add one more keyword in the paper.
Response- Keywords: Swarm Robotics, Optimization, Multi Robotic Systems, Evolutionary Algorithms, Simulation
One keyword changed (Algorithm à Evolutionary algorithm) and one keyword added: Simulation
- The references should be cited in the Introduction part, and also give some descriptions of them.
Response- 18 references are cited in the Introduction part. There are 16 new references are added; Ref. [3] - [18]
- Please fix some figures with minor typoes such as Fig. 2 and Fig. 3.
Response- The figures mentioned were fixed and scaled.
- The authors are suggested to present a pseudocode or procedure for the proposed algorithm in the paper.
Response- The pseudocode of the study is given in Fig. 17 at section 3.1.3.
- The reference part is suggested to be updated with some other applications on evolutionary algorithms to enhance related and previous work part, e.g., Three-dimensional route planning based on the beetle swarm optimization algorithm. A novel convolutional neural network model based on beetle antennae search optimization algorithm for computerized tomography diagnosis.
Response- Paragraph 6. and 8. were added to the Introduction part. The paragraph 6 is about different evolutionary optimization techniques examples on swarm robotics in the literature, in the paragraph 8 is given application examples of swarm robotics in the literature. The articles mentioned were added to the Introduction part and Reference part.
Reviewer 2 Report
This manuscript summarized the features and classification of swarm robotics through the comparison of parameters between different multi-robot systems. In particular, the communication methods and key technologies are described in detail, several optimization methods inspired by nature and problems caused by optimization design are explained. Throughout the manuscript, the most innovative aspect described by authors is the simulation study for swarm robots. There are some changes and explanations that should be provided before consideration of acceptance.
- The perception range of an individual in a swarm robot is one of the mostimportant parameters. Excessive sensing radius will increase the cost and information honor, and smaller perception range will lead to the disconnection of network nodes. How to weigh or choose this range of perception after being inspired by biological evolution?
- As we know, due to the simulation software will basically provide users with ideal boundaries and conditions, there will be some gaps between simulation and practice. Forthe work of section 3.1.3, what are the relevant problems caused by the optimization design and communication to focus on?
- The manuscript lists some optimization methods derived from nature, which can be used in swarm robotics to accomplish certain tasks.What can swarm robots do with these optimization algorithms in real life? Please list 2-3 scenarios, which should correspond to the optimization algorithms. This can highlight the role and significance of this work in the field of swarm robots.
- The manuscript requires some editing, particularly for language. For example, “features of swarm robotics and their classification from different angles will be mentioned”. It is more appropriate to change “angles” to “perspectives”.
Author Response
Thank you very much for your valuable suggestions. We have addressed them during the revision of the manuscript.
- The perception range of an individual in a swarm robot is one of the most important parameters. Excessive sensing radius will increase the cost and information honor, and smaller perception range will lead to the disconnection of network nodes. How to weigh or choose this range of perception after being inspired by biological evolution?
Response- Considering that living things change their genes and characteristics over time in nature according to population and environment, such changes are inevitable in the design of swarm systems. For example, it is possible to expand or reduce the communication perception range in software or hardware in the next study by considering the width of the area where the swarm is located, the size of the swarm, the presence of objects that prevent communication in the environment, the lack of communication tools, battery conditions. Section 3.1.3 has been handled only in software in the simulation environment, and in case of communication interruption, the setRange value is adjusted to ensure that the communication of the robots is not interrupted. However, in the next study, it is possible to assign this task to some special robots in the swarm by leaving immobile the robots with low battery at certain points and acting as repeaters or creating heterogeneous swarms. In the opposite scenario, the signal level will be decreased with the algorithm of swarm structures for the areas communication not interrupted by objects.
- As we know, due to the simulation software will basically provide users with ideal boundaries and conditions, there will be some gaps between simulation and practice. Forthe work of section 3.1.3, what are the relevant problems caused by the optimization design and communication to focus on?
Response- In Section 3.1.3, the modeling of the PSO suitable for the characteristics of swarm robots [1] is done on all individuals in a distributed architecture and provides the individual movements independently but in harmony with the swarm using the swarm software. Additionally, a study was carried out to include the problems mentioned in sections 3.1.1 and 3.1.2 in optimization. Thus, coordinate conflicts and unnecessary behaviors in the control of the swarm have been eliminated, and solutions that will eliminate communication interruption or signal echoes have been applied in swarm robotics that move using PSO algorithm. In newly added Figure 17, this structure is given in the pseudocode.
- The manuscript lists some optimization methods derived from nature, which can be used in swarm robotics to accomplish certain tasks. What can swarm robots do with these optimization algorithms in real life? Please list 2-3 scenarios, which should correspond to the optimization algorithms. This can highlight the role and significance of this work in the field of swarm robots.
Response- The new section -Section 2.5. Optimizing in swarm robots; Real life examples- about real life examples was created. Four examples using swarm robots in the literature were added.
- The manuscript requires some editing, particularly for language. For example, “features of swarm robotics and their classification from different angles will be mentioned”. It is more appropriate to change “angles” to “perspectives”
Response- The requested change has been made.
Reviewer 3 Report
The article is relevant, it considers the general structure of swarm robotics. The article presents the algorithms that form the basis of swarm robotics and develops optimization methods used in swarm control.
Author Response
- The article is relevant, it considers the general structure of swarm robotics. The article presents the algorithms that form the basis of swarm robotics and develops optimization methods used in swarm control.
Response-
Thank you so much for your comment.
Reviewer 4 Report
Swarm robotics as a research topic can not been directly related with sensor problems. The survey on swarm, (perhaps, the unique positive value of the paper) is quite limited. In fact, it can be seen as a bibliographic work (limited) because the practical work is a simple exercise use of webots software. It is not realistic. At own experience the main practical issues with odometry are not considered, and the execution time is not clearly shown, and it is known that the performance is highly degradated as time evolves. But the biggest problem comes from the fact that sensorial concepts that very important in robotics are not considered.
Author Response
- Swarm robotics as a research topic can not been directly related with sensor problems. The survey on swarm, (perhaps, the unique positive value of the paper) is quite limited. In fact, it can be seen as a bibliographic work (limited) because the practical work is a simple exercise use of webots software. It is not realistic. At own experience the main practical issues with odometry are not considered, and the execution time is not clearly shown, and it is known that the performance is highly degradated as time evolves. But the biggest problem comes from the fact that sensorial concepts that very important in robotics are not considered.
Response- According to the valuable suggestions “survey on swarm, (perhaps, the unique positive value of the paper) is quite limited” Section 2.2.2 was enhanced. Including articles related to sensorial concept and sensor problems in swarm robotics have been added to the Introduction part and the References.
Round 2
Reviewer 1 Report
The authors have addressed all comments well. This paper can be accepted.
Author Response
Dear Reviewer,
As suggested the whole manuscript has been revised from the aspect of the grammar and syntax and necessary changes were made. Especially, the Abstract, the Discussion and Conclusions parts have been re-written to improve the reader’s appeal. The authors sincerely thanks to the reviewers and the editors for their valuable contributions to the manuscript. Without your efforts, the article would not have reached this advanced level.
Sincerely yours.
Reviewer 2 Report
The authors have addressed all the concerns and carefully revised the manuscript according to the reviewer's comments. In my opinion, the present revision can be accepted.
Author Response

(The authors gave the same response as above.)
